# Validity of Instrumented Insoles for Step Counting, Posture and Activity Recognition: A Systematic Review

**DOI:** 10.3390/s19112438

**Published:** 2019-05-28

**Authors:** Armelle M. Ngueleu, Andréanne K. Blanchette, Désirée Maltais, Hélène Moffet, Bradford J. McFadyen, Laurent Bouyer, Charles S. Batcho

**Affiliations:** 1Centre for Interdisciplinary Research in Rehabilitation and Social Integration (CIRRIS), Centre intégré universitaire de santé et de services sociaux de la Capitale-Nationale (CIUSSS-CN), Quebec City, QC G1M2S8, Canada; armelle.ngueleu.1@ulaval.ca (A.M.N.); andreanne.blanchette@fmed.ulaval.ca (A.K.B.); Desiree.Maltais@rea.ulaval.ca (D.M.); Helene.Moffet@rea.ulaval.ca (H.M.); Brad.McFadyen@fmed.ulaval.ca (B.J.M.); Laurent.Bouyer@rea.ulaval.ca (L.B.); 2Department of Rehabilitation, Faculty of Medicine, Université Laval, Quebec City, QC G1M2S8, Canada

**Keywords:** insoles, criterion validity, posture and activity recognition, step counting

## Abstract

With the growing interest in daily activity monitoring, several insole designs have been developed to identify postures, detect activities, and count steps. However, the validity of these devices is not clearly established. The aim of this systematic review was to synthesize the available information on the criterion validity of instrumented insoles in detecting postures activities and steps. The literature search through six databases led to 33 articles that met inclusion criteria. These studies evaluated 17 different insole models and involved 290 participants from 16 to 75 years old. Criterion validity was assessed using six statistical indicators. For posture and activity recognition, accuracy varied from 75.0% to 100%, precision from 65.8% to 100%, specificity from 98.1% to 100%, sensitivity from 73.0% to 100%, and identification rate from 66.2% to 100%. For step counting, accuracies were very high (94.8% to 100%). Across studies, different postures and activities were assessed using different criterion validity indicators, leading to heterogeneous results. Instrumented insoles appeared to be highly accurate for steps counting. However, measurement properties were variable for posture and activity recognition. These findings call for a standardized methodology to investigate the measurement properties of such devices.

## 1. Introduction

There is growing evidence regarding the role of regular physical activity in the improvement and preservation of functional autonomy and in the prevention of many diseases and disorders [1,2,3,4,5]. For example, it has been shown that regular practice of physical activity contributes in preventing recurrent stroke [1,5], obesity [1,4], cardiovascular diseases [1,2,3], and cancer [1]. Physical activity can be defined as any bodily movement produced by skeletal muscles resulting in energy expenditure [6]. The identification in daily life of relevant postures (e.g., sitting, standing) and activities (e.g., walking, jogging, descending/ascending stairs or ramp, cycling) provides important information regarding individuals’ active or sedentary behavior and is thus a crucial component of daily physical activity measurement.

Physical activity may be evaluated using subjective and objective methods [7,8,9,10,11,12,13,14]. Subjective methods such as questionnaires [7,8] and individual diaries [15] are relatively inexpensive and are the more feasible method in large population-based studies. However, they present some limitations such as recall error and risk of over-reporting physical activity [16,17]. Objective methods include a wide variety of commercially available body-worn devices to detect movement or indirectly measure energy expenditure such as pedometers, accelerometers and heart rate monitors [14,15]. Many recent studies have used these objective methods to quantify physical activity in different populations (e.g., those without disease or disability, stroke survivors, people with Parkinson disease, traumatic brain injury or cerebral palsy and those who have undergone an amputation) [7,18,19,20]. The criterion validity of these objective methods varies largely from one device to another [7,21,22] with multi-sensor activity monitors (e.g., Step Activity Monitor, GaitUp Physiolog, etc.) showing better psychometric properties [23,24,25]. However, not all of these devices are appropriate for daily use in people with disability. In fact, one study, for example, evaluated the usability of seven activity monitors in older adults based on four criteria: (a) ease in applying the monitor, (b) ease in reading the step display, (c) comfort, and (d) ease in accessing step display on the monitor [22]. This study concluded that common barriers to activity monitoring were an inability to apply the monitor and a difficulty in accessing step display on the monitor [22].

Smart insoles, equipped with miniaturized sensors appear as a potential solution for unobtrusive monitoring of daily activities [23,24,25,26], given the fact that humans wear shoes for many hours a day and present microelectromechanical systems have enabled instrumentation of insoles. Moreover, it has been suggested that instrumented insoles may be less expensive than available activity monitors [27].

A recent systematic review assessed the available information on the psychometric properties (mainly criterion validity) of different activity monitors in stroke survivors [7]. However, only one smart insole was included in this review and no conclusion was reported regarding the criterion validity of smart insoles as an activity monitor. The authors did report that smart insoles were unobtrusive, lightweight and comfortable, and thus they were potentially user-friendly with high acceptability.

Another comparative review summarized the development of footwear-based systems and their applications [20]. This review reported some accuracies of footwear-based systems for gait monitoring, plantar pressure measurement, posture and activity recognition, body weight and energy expenditure estimation, biofeedback, navigation, and fall risk applications in individuals with and without physical limitations such as stroke survivors and people with Parkinson disease or cerebral palsy. However, the analysis of psychometric properties of insoles for physical activity monitoring was not the study’s main focus. Given the growing interest in the use of smart insoles as a monitoring device, and the importance of physical activity, there is a need to investigate the measurement properties of smart insoles as applied to physical activity monitoring. The most important measurement property in this context is validity, which can be quantified in terms of accuracy, precision, specificity, sensitivity, measurement error and criterion validity. In assessing criterion validity in this context, the agreement between step count, posture and activity as measured using instrumented insoles and by direct observation should be determined.

The aim of the present systematic review was to assess and synthesize the available information on the criterion validity of the instrumented insoles in identifying postures, activities and the number of steps.

## 2. Methods

### 2.1. Data Source and Search Strategy

This systematic review followed the Preferred Reporting Items for Systematic Reviews and Meta-Analyses (PRISMA) guidelines [28]. Six electronic databases (MEDLINE, EMBASE, IEEE Xplore, Cochrane Library, Scopus and Web of Sciences) were searched for studies reporting criterion validity information from instrumented insoles that related to step counting, posture and activity recognition. Free and indexed search terms were used following four steps as follows: (#1) measurement system (search terms related to insole), (#2) measurement properties (search terms related to criterion validity) and (#3) outcomes (search terms related to step count, posture and activity recognition). For the final step, search parameters #1, #2 and #3 were combined to retrieve references that covered all three concepts (#4). The search strategy was adapted for each database and limited to English or French language articles published from a given database’s inception to 6 May 2019. The detailed search strategy is presented in Table 1.

### 2.2. Selection Criteria

We considered for inclusion only studies assessing the psychometric properties of instrumented insoles to quantify step counting and to detect posture and activity recognition. Instrumented insoles were defined as insoles integrating at least one of the following: pressure sensors, an accelerometer, a gyroscope, a magnetometer, an inertial measurement unit (IMU), or other electronic sensors (e.g., heart rate sensors). Outcomes included the quantification of stride or step count, recognition of posture (lying, sitting and standing), or activities (walking, jogging, ascend/descend ramp or stairs, cycling and elevator up/down). The targeted psychometric property was criterion validity. Studies that involved another measurement system in addition to insoles were included if data of the instrumented insole could be extracted. Papers were included if they were scientific papers with available full-text. 

### 2.3. Article Selection

After removing duplicate references from the search results, two reviewers independently screened the titles and abstracts to identify potentially eligible articles based on the selection criteria. Preliminary selection results were compared and discrepancies were resolved by discussions between the reviewers. If it was unclear based on the title and abstract whether or not a publication met the selection criteria, the full-text of the article was read before a final decision was made. The full-text articles of all pre-selected references were independently reviewed by the two reviewers to determine if articles met the selection criteria. Discrepancies were again discussed and, in case of no consensus, a third reviewer was consulted for final decision regarding the selection. 

### 2.4. Data Extraction

The relevant data from the selected full-text articles were extracted by one reviewer. Extracted data were as follows: full article reference, participants’ characteristics (diagnosis, age), data collection setting and duration, insole design (sensing element, sampling frequency, data transmission method), criterion methods (or gold standard), algorithms used, outcomes (step count, postures and activities), validity.

### 2.5. Methodological Quality Assessment

The methodological quality of the studies reported in the selected articles was assessed using a structured quality appraisal tool developed by MacDermid [29]. This tool consists of 12 criteria pertaining to the study question and design, measurement methods, analyses and recommendations. Each item was scored as 0, 1, 2 or NA (not applicable) giving a maximum possible score of 24. For each article, the quality score was expressed as a percentage calculated as:(1)$$\mathrm{Quality}\;\mathrm{score}\;=\;\frac{\mathrm{obtained}\;\mathrm{score}}{\mathrm{total}\;\mathrm{possible}\;\mathrm{score}}\;\times \;100\%$$

As per de Oliveira et al. [30], study quality was categorized as follows: “high quality” ≥ 80.0%, “good quality” between 70.0% and 79.9%, “moderate quality” for scores between 50.0% and 69.9%, and “low quality” representing scores < 50.0%. These categories and scores were used to assist with interpretation of the review findings. No article was omitted from the review based study quality, however.

## 3. Results

The electronic search retrieved 2030 records, from which 930 duplicates were removed. The title and abstract of the remaining 1100 records were screened (1015 were removed) and then there was a review of the full-text of the remaining 85 references (52 removed). Thirty-two articles (with independent studies) met the inclusion criteria. The search history and selection process are presented in Figure 1. From the 33 included articles, 27 reported on studies that evaluated posture and activity recognition while step counting was reported in seven of them. One article [31] reported on step counting as well as on posture and activity recognition. Most of the included articles were published in the past 10 years as illustrated in Figure 2. Table 2 and Table 3 present the description of technical features of insoles respectively for posture and activity recognition, and step count. Table 4 summarizes the criterion validity of insoles for posture and activity recognition, while criterion validity of insoles for step count is included in Table 3. Finally, Table 5 presents the summary of methodological quality appraisal of included studies using MacDermid.

### 3.1. Insole Models and Technical Features

The 33 articles described 17 different insole models, most of them (16/17) being academic research prototypes (see Table 2 and Table 3). Only one of these 17 insole models was commercially available [32,33]. Data transmission methods were Bluetooth, wireless and wire modules with sampling frequencies varying from 10 Hz to 400 Hz (see Table 2 and Table 3). For step detection, instrumented insoles were validated using visual observation [25,31,34,35], other devices (the Runtastic pedometer application and other smartphone applications) [34,36], or using a predefined number of steps [24,36,37] (see Table 3). To validate the instrumented insoles for posture and activity recognition, comparisons were made between the smart insole data and that collected from direct observation during data collection or from a video recording or from other wearable devices (2D accelerometer (ADXL202), gyroscope (Murata, ENC-03J), ActivPAL device, PPAC (plantar-pressure based ambulatory classification) and FF (foot force sensor) + GPS [18,26,31,32,33,38,39,40,41,42,43,44,45,46,47,48,49,50,51,52,53,54,55,56,57,58,59,60] (see Table 2).

### 3.2. Populations

Most of the selected articles (n = 26) reported on studies that assessed the measurement properties of instrumented insoles in healthy adults only [20,26,35,38,39,43,44,45,46,47,48,49,50,51,52,53,54,55,57,59,60] (see Table 2 and Table 3). Three articles reported this information in stroke survivors (51.7 ± 45.1 and 65.2 ± 41.8 months post-stroke) [31,40,42]. Two articles reported on older adults [32,33], one on both older and younger adults [41], and one on both young healthy adults and persons with an amputation [56]. A total of 290 participants were included in the selected studies (n = 33), with sample sizes ranging from 1 to 40. Participants were aged between 16 and 75 years old (see Table 2 and Table 3).

### 3.3. Study Settings and Test Durations

For posture and activity recognition, insoles were evaluated between 1 min and 13 h in a laboratory setting (n = 15) [26,31,40,42,43,44,45,46,47,48,49,50,53,58,60], as well as within the community, including outdoor activities (n = 8) as illustrated in Table 2 [26,32,33,41,43,48,55,60]. In eight articles, the setting was not mentioned [38,39,51,52,54,56,57,59]. For step counting, evaluations were also performed in both laboratory (n = 3) [25,31,35,37] and community (n = 2) [24,34] environments, but for shorter durations (2 min, [31] and 6 minutes [35]), on predefined distances of 16 meters [25] and 720 m [34] or for a predetermined number of steps (50 steps [36] and 100 steps [24,37]) as illustrated in Table 3.

### 3.4. Algorithms

Different algorithms were used to identify postures and detect activities, including a decision tree (DT, n = 6), linear discriminant analysis (LDA, n = 3), multinomial logistic discrimination (MLD, n = 4), a convolutional neural network (CNN, n = 1), a support vector machine (SVM, n = 8), multi-layer perceptron (MLP, n = 2), an artificial neural network (ANN, n=3), k-nearest-neighbors (KNN, n = 2) and other methods (n = 2) as illustrated in Table 2. For step counting, algorithms used were based on the sum of pressure signals (n = 1), average of pressure signals (n = 2) and acceleration variance (n = 1) as illustrated in Table 3.

### 3.5. Outcomes

Regarding types of posture and activity, instrumented insoles were used to identify when participants were sitting [26,31,32,33,40,42,43,44,45,46,47,48,49,50,51,52,53,54,56,57,58], standing [26,31,32,33,38,39,40,42,43,44,45,46,47,48,49,50,51,52,53,54,56,57,58,59], walking [26,31,32,33,38,39,40,42,43,44,45,46,48,49,50,51,52,53,54,55,56,57,58,59,60], running [41,48,49,51,52,53,54,57,59], jumping [43,57,59], cycling [26,42,44,48,49,51,53,54,55,58,59], ascending/descending a ramp [32,39,60] or stairs [26,32,33,39,41,42,47,49,51,53,54,56,57,60], car driving [26,55], vacuuming [26], taking an elevator up/down [32,33], dancing [43], lying down [26,44,57], shelving items [26], washing [26], sweeping [26], or falling down [50] (see Table 4).

### 3.6. Criterion Validity of Instrumented Insoles

For posture and activity recognition, criterion validity was assessed using accuracy, precision, specificity, sensitivity and identification rate (see Table 4). Overall accuracies varied from 75.0% to 100% for posture and activity recognition, as reported in 22 articles [26,32,33,39,40,42,43,44,45,46,47,48,49,51,52,53,55,56,57,58,60]. Precision, reported in 10 articles [31,32,33,42,44,48,49,53,59,60], varied from 65.8% to 100%. In two articles, precision of instrumented insoles varied from 3.0% to 27.2% for ascending/descending stairs [42,53].

Only four articles reported an assessment of the specificity of posture and activity recognition with values ranging from 98.1% to 100% [32,33,52,57], except for ascending/descending stairs for which specificity varied from 43.1% to 97.4% [52]. Sensitivity was reported in 13 studies and varied between 73.0% and 100% [31,32,33,40,42,44,48,49,52,53,57,59,60], except for down/upstairs (between 10.0% and 91.7%) [42,49,52,53,57]. Identification rate was reported in three studies and varied from 66.2% to 100% [38,41,50]. In one article, the instrumented insole was reported to identify sitting with an identification rate varying from 0% to 98.0% [50]. Accuracy was the most widely used index to assess criterion validity of insoles for posture and activity recognition (see Table 4). Only two articles reported accuracy, precision, specificity and sensitivity [32,33].

For step counting, accuracies were from 94.8% to 100% (see Table 3) [24,25,35]. Similarly, one article reported an intraclass correlation coefficient (ICC) of 0.99 for number of steps [31]. Error rates of 4% and 0% were reported for walking and running respectively [36]. In two articles, the reported measurement errors were 0% [34] and < 1% [37] for step counting during walking. The methods used to validate step counting were different from one article to another.

### 3.7. Methodological Quality of Included Articles

Based on MacDermid criteria, the total methodological quality scores for each reported article were calculated (Table 5). Total quality scores varied between 12 and 23 points corresponding to 50.0% and 95.6%. For posture and activity recognition, the methodological quality was high in 16 articles, good in six articles, and moderate in four articles as illustrated in Table 5. For step counting, methodological quality was high for one article, good for two articles and moderate for three articles (see Table 5).

## 4. Discussion

The aim of the present review was to assess and synthesize the available information on the criterion validity of instrumented insoles in detecting posture, activity and steps. This systematic review included 33 articles that reported smart insole criterion validity data such as the accuracy, precision, specificity, sensitivity, identification rate, and measurement error for step counting, posture and activity recognition. Accuracy varied from 75.0% to 100%, precision from 65.8% to 100% and specificity from 98.1% to 100%. These values excluded the detection of ascending/descending stairs because in some articles, the criterion validity of this activity was reported to be very low (from 3.0% to 53.0%) [26,42,52] except in studies of Sazonov et al. [49], Zhang et al. [41], Sugimoto et al. [51], Peng et al. [47], and Chen et al. [39]. Sensitivity varied from 73.0% to 100%, identification rate from 66.2% to 100% and measurement error was of 4.0%. Walking, standing and sitting were the most frequently assessed activities. In summary, the criterion validity of instrumented insole varied from one article to another, and was expressed by different indicators. Overall, instrumented insoles appear to be best at monitoring of steps.

The variation in the reported instrumented insole criterion validity results could be related to several factors such as methods of training algorithms, dataset size, and heterogeneity of activities and postures. For example, larger datasets may result in a higher rate of successful classification. The training and validation algorithms from Edgar et al., [43] were based on datasets of 4800 and 2400 feature vectors respectively, which yielded a successful classification of 99.3% for training and 89.6% for validation. Zhang et al, [41] on the other hand, reported an identification rate of 98.8% for training and 98.3% for validation based on 11 268 feature vectors for training and 8687 for validation. The results from Sazonov et al. [42,61] suggest that the algorithm training method may also influence the accuracy of events detection results. These researchers used two models of training algorithms, an individual model and a group model. The individual model, a training algorithm for each individual participant, yielded higher accuracy for both training and validation (99.9% and 99.1%, respectively) than the group model (95.0% and 91.5%, respectively) [40,42]. Finally, the variability of reported accuracies between studies may be due to the heterogeneity of experimental conditions regarding activities to be detected (sitting, walking, etc.). Indeed, with an experimental task of only 3 distinct activities (standing, sitting and walking)*,* articles had reported high criterion validity [31,40,45,46] while in other articles, experimental conditions with more than 3 selected activities (standing, sitting, walking, car driving, vacuuming, ascending/descending stairs, elevator up/down, dancing, lying down, shelving items, washing, etc.) led to lower detection rate [32,33,39,43,44,53,57]. These differences in the experimental conditions do not allow comparisons of the different models of instrumented insoles that were used across studies. It is, therefore, difficult to state which of the insoles was best. To make such a conclusion, more consistent studies are needed that would have tested different models of instrumented insoles through standardized experimental conditions. However, the findings of this systematic review may help identifying the most appropriate device for a given application.

Some authors have reported the inability of their algorithms to discriminate between a walking activity and standing posture, with confusion occurring mainly in people with low walking speed (0.69 ± 0.35 m/s) such as older adults and stroke survivors [32,40]. This indicates that event detection algorithms may be sensitive to the amplitude of movements, so that the detection of upright events like standing or slow walking based on sensor signals can be confused, leading to false positive results. Such observations have been reported in the literature with other physical activity monitoring devices (pedometers, accelerometers and inertial sensors) that are known to be less accurate in detecting activities at slow walking speed [22,62,63,64]. However, it worth noting that among the 33 articles included in this review, only three reported an assessment of the criterion validity of instrumented insoles in stroke survivors [31,40,42] and only two in older adults [32,33]. Therefore, more studies are required to investigate the accuracy of instrumented insoles in discriminating standing from walking at a slow speed.

Ten different algorithms were identified in this systematic review of 33 articles. These varied from simple (for example, binary decision trees) to complex machine learning algorithms (for example, Support Vector Machine, SVM). It is difficult to directly compare these algorithms of the variability in the experimental conditions under which they were used. However, two articles examined the accuracy of the same instrumented insole for posture and activity recognition using either two (SVM, MLP) [53] or three algorithms (SVM, MLP, MLD) [48] under the same experimental condition and on the same dataset. The authors [48,53] concluded that there was no significant difference between these algorithms. However, the storage space requirement was high for SVM compared to MLD and MLP [48]. MLD and MLP can run on wearable devices such as insoles, while SVM runs only on a computer, and data are then stored and processed off-line. Thus, SVM cannot be used for real-time posture and activity recognition.

This systematic review highlights the need for a consensus on the methodology and the measurement quality to consider (e.g., accuracy, precision, specificity) when validating insoles for posture and activity recognition. The ideal study design would be the one that compares different models of insoles based on the same experimental protocol. However, this seems complex since, contrary to other wearable devices such as wrist-worn accelerometers, one participant can only wear a maximum of two insoles at a time. Therefore, in place of multiple comparisons of insoles within the same data collection, standardized methodology of insole validation would be useful for future research. Consensus on postures and activities that should be included in experimental designs for insole validation studies would also facilitate the comparison of psychometric properties for different models of this new monitoring device. Moreover, the use of similar methods of evaluation and standardized postures and activities could enable pooling data for comparative analyses [7].

Given that most of the studies in this review were conducted in a laboratory setting, we found limited evidence regarding the criterion validity of insoles within an outdoor context. Future work should consider evaluating the psychometric properties of insoles in the community setting. To make this possible, sensors and circuit boards should be integrated into insoles rather than having separate components that need to be connected for data collection, as was the case for most of the 16 insole models evaluated here. With external components to carry on during data collection, the use of some insoles may be uncomfortable for the users while performing daily activities. Another limitation of the findings from this review is that many studies enrolled only young, healthy adults. Such samples do not allow testing the insoles in individuals with various or unstable walking patterns. Indeed, accuracy of instrumented insoles could vary from a normal walking pattern to pathological or modified walking patterns for step counts, posture and activity recognition. Thus, there is a need to evaluate insoles on different walking patterns.

## 5. Conclusions

This systematic review provides a summary of the validity of instrumented insoles for steps count and activity recognition. Instrumented insoles appeared to be highly valid for step counting; but measurement qualities were variable for posture and activity recognition due to the heterogeneity of experimental conditions and tasks on which different models of insole were tested using ten different algorithms. The most frequently assessed activities were walking, standing and sitting. In addition, several indicators (e.g. accuracy, precision, measurement error, etc.) were used to assess criterion validity of the insoles. Further research should standardize indicators of criterion validity to be considered, and the experimental postures and activities used for testing the insoles.

## Figures and Tables

**Figure 1 sensors-19-02438-f001:**
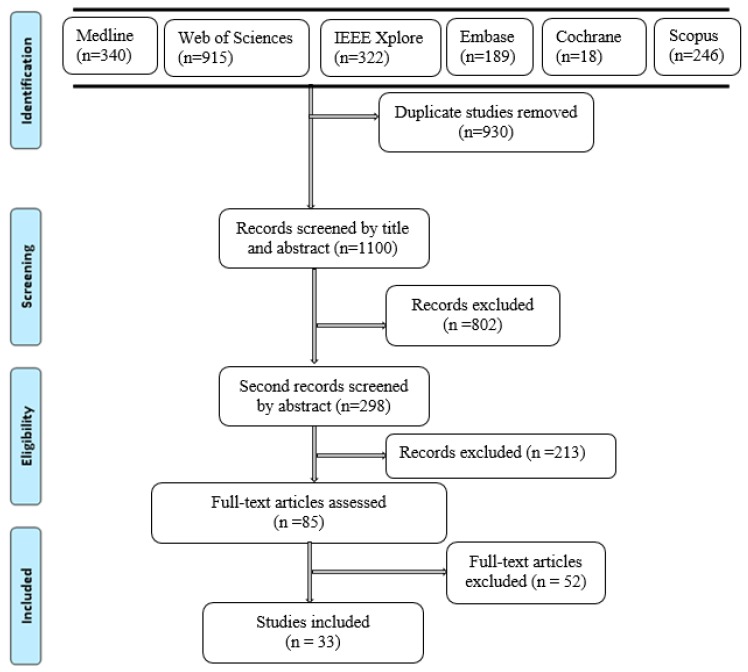
Flowchart of articles identification and selection process according to PRISMA guidelines.

**Figure 2 sensors-19-02438-f002:**
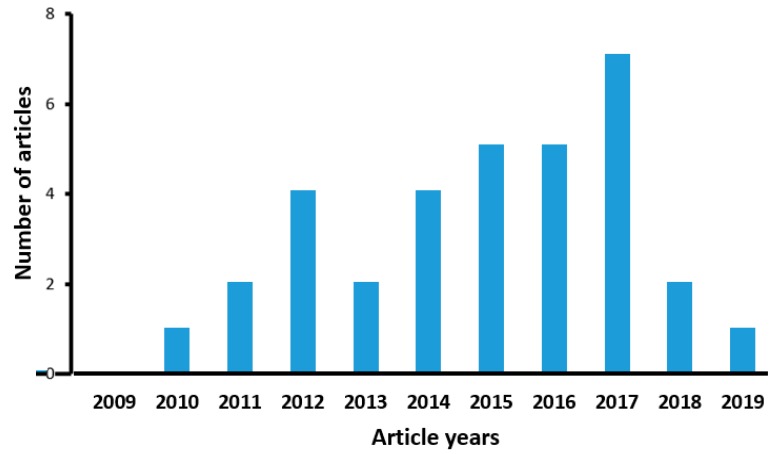
Number of articles having reported the criterion validity of instrumented insoles in the past 10 years, for steps, activity and posture detection.

**Table 1 sensors-19-02438-t001:** Detailed literature search strategy.

**Medline via OVID Search Strategy**	
**#1**	Insole* OR foot orthos* OR instrument* shoe* OR smartshoe* OR shoe* plantar pressure OR feet orthos*: ti,ab,kw OR exp foot orthoses/
**#2**	psychometric qualit* OR psychometric propert* OR validit* OR measurement error* OR specificit* OR precision OR accura* OR sensibilit: ti,ab,kw OR exp psychometric quality/ OR exp psychometric property/ OR exp criterion validity/ OR exp accuracy/
**#3**	Posture* OR activit* OR classif* OR step count* OR stride count* OR number of step*: ti,ab,kw OR exp posture/ OR exp step count/ OR exp stride count/ OR exp number of step/
**#4**	#1 AND #2 AND #3
**Medline via PubMed Search Strategy**	
**#1**	foot orthos*[mh] OR instrument*[mh] shoe*[mh] OR smartshoe*[mh] OR shoe* plantar pressure[mh] OR feet orthos*[mh]
**#2**	psychometric qualit*[mh] OR psychometric propert*[mh] OR validit*[mh] OR measurement error*[mh] OR specificit*[mh] OR precision[mh] OR accura*[mh] OR sensibilit*[mh]
**#3**	Posture*[mh] OR activit*[mh] OR classif*[mh] OR* step count*[mh] OR stride count*[mh] OR number of step*[mh]
**#4**	#1 AND #2 AND #3
**Embase Search Strategy**	
**#1**	’insole*’:ti,ab,kw OR ’foot orthos*’:ti,ab,kw OR ’instrument* shoe*’:ti,ab,kw OR ’smart shoe*’:ti,ab,kw OR ’shoe* plantar pressure’:ti,ab,kw OR ’feet orthos*’:ti,ab,kw OR ’foot orthosis’/exp OR ’insole’/exp OR ’instrumented shoe’/exp
**#2**	’psychometric qualit*’:ti,ab,kw OR ’psychometric propert*’:ti,ab,kw OR ’validit*’:ti,ab,kw OR ’specificit*’:ti,ab,kw OR ’precision’:ti,ab,kw OR ’accura*’:ti,ab,kw OR ’sensibilit*:ti,ab,kw OR ’psychometric quality’/exp OR ’psychometric property’/exp OR ‘criterion validity’/exp OR ‘specificity’/exp OR ’precision’/exp OR ‘accuracy’/exp OR ’sensibility’/exp
**#3**	Posture: ti,ab,kw OR activit*: ti,ab,kw OR classif*: ti,ab,kw OR* step count*: ti,ab,kw OR stride count*: ti,ab,kw OR number of step*: ti,ab,kw OR ‘posture’/exp OR ‘activity’/exp OR ‘number of step’/exp OR ‘step count’/exp
**#4**	#1 AND #2 AND #3
**IEEE Xplore Search Strategy**	
**#1**	Insole OR foot orthosis OR instrumented shoe OR smartshoe OR shoe plantar pressure OR feet orthosis
**#2**	Psychometric quality OR psychometric property OR validity OR measurement error OR specificity OR precision OR accuracy OR sensibility
**#3**	Posture OR activity OR classification OR step count OR stride count OR number of step
**#4**	# 1 AND #2 AND #3
**Web of Science Search Strategy**	
**#1**	Insole* OR foot orthos* OR instrument* shoe* OR smart shoe* OR shoe* plantar pressure OR feet orthos*:ti,ab,kw
**#2**	Psychometric qualit* OR psychometric propert* OR validit* OR measurement error* OR specificit* OR precision OR accura* OR sensibilit*:ti,ab,kw
**#3**	Posture OR activit* OR classification OR step count* OR stride count* OR number of step*:ti,ab,kw
**#4**	#1 AND #2 AND #3
**Cochrane Search Strategy**	
**#1**	Foot orthos OR instrument* shoe* OR smartshoe* OR shoe* plantar pressure OR feet orthos*:ti,ab,kw
**#2**	Psychometric qualit* OR psychometric propert* OR validit* OR measurement error* OR specificit* OR precision OR accura* OR sensibilit*:ti,ab,kw
**#3**	Posture OR activit* OR classification OR step count* OR stride count* OR number of step*:ti,ab,kw
**#4**	#1 AND #2 AND #3
**Scopus Search Strategy**	
**#1**	Foot orthosis OR instrumented shoe OR smartshoe OR shoe plantar pressure OR feet orthosis OR insole
**#2**	Psychometric quality OR psychometric property OR validity OR measurement error OR specificity OR precision OR accuracy OR sensibility
**#3**	Posture OR activity OR classification OR step count OR stride count OR number of step
**#4**	(Foot orthosis OR instrumented shoe OR smartshoe OR shoe plantar pressure OR feet orthosis OR insole) AND (psychometric quality OR psychometric property OR validity OR measurement error OR specificity OR precision OR accuracy OR sensibility) AND (posture OR activity OR classification OR step count OR stride count OR number of step)

* represents a truncation that allows to develop all derived forms of a word.

**Table 2 sensors-19-02438-t002:** Description of technical features of insoles for posture and activity recognition.

	Sensing Elements	Sampling Frequency (Hz)	Data Transmission Methods	Populations	Age (Years)	Algorithms	Comparison Methods	Settings	Test Durations
Hegde et al. [26]	3 FSRs 402, ACC.	50	Bluetooth	15 adults	M: 26.6 (3.4) F: 23.3 (5)	MLD	ActivPAL device	Laboratory and community	1 h (laboratory), 8 AM–9 PM (free-living)
Fulk et al. [31]	5 FSRs, ACC.	400	Bluetooth	12 stroke survivors	62.1 (8.2)	ANN	Video record	Laboratory	2 min/activity
Achkar et al. [32]	8 FSRs, IMU*, barometer	200	Wire connection	10 older adults	65–75	DT	ACC., gyroscope	Community	1 h/participant
Achkar et al. [33]	8 FSRs, IMU*, barometer	200	Wire connection	10 older adults	69.9 (3.1)	DT	2D ACC., gyroscope	Community	4 h (total)
Anlauff et al. [38]	4 FSRs	200	Bluetooth	8 adults	25.2 (NA)	NA	Visual observation	NA	45 min (total)
Chen et al. [39]	4 FlexiForce, IMUs*	100	Wireless mode	7 adults	24.1 (0.5)	LDA	Visual observation	NA	15 min/experiment
Fulk et al. [40]	5 FSRs, Acc.	25	Wireless link	8 stroke survivors	60.1 (9.9)	SVM	Visual observation	Laboratory	1 min/activity
Zhang et al. [41]	32 miniature pressure sensors	32	Wire connection	40 adults	27.3 (13.2)	ANN	Visual observation	Outside, laboratory	50 m
Zhang et al. [42]	5 FSRs, ACC.	25	Bluetooth	12 stroke survivors	62.1 (8.2)	DT	Visual observation	Laboratory	2 min/activity
Edgar et al. [43]	3 pressure sensors, ACC.	100	Bluetooth	1 adult	22	ANN	Visual observation	Indoor and outdoor	3 min/activity
Hegde et al. [44]	2 or 3 FSRs 402	NA	Bluetooth	3 adults	24 (4.5)	MLD	Visual observation	Laboratory	20 min/activity
Lin et al. [45]	48 pressure sensors, IMU	100	Bluetooth	8 people	NA	KNN	Visual observation	Indoor	NA
Lin et al. [46]	48 pressure sensors, IMU	100	Bluetooth	8 people	NA	NA	Visual observation	Indoor	10 trials/participant
Peng et al. [47]	7 FSR402	25	Wireless module	1 adult	24	SVM	Visual observation	Indoor	NA
Sazonov et al. [48]	5 FSRs, ACC.	400	Bluetooth	19 adults	28.1 (6.9)	SVM, MLP, MLD	Video	Indoor and free-living	52.5 h (total)
Sazonov et al. [49]	5 FSRs, ACC.	25	Wireless module	9 adults	23.7 (4.3)	SVM	Visual observation	Laboratory	11 h 36 min (total)
Shang et al. [50]	2 pressure sensors, ACC.	NA	Wireless module	3 adults	NA	Threshold method	Visual observation	Laboratory	NA
Sugimoto et al. [51]	7 pressure sensors	20	USB port	2 adults	NA	LDA	Visual observation	NA	2 min
Tang et al. [52]	5 FSRs, ACC.	25	Wireless module	9 adults	23.6 (4.3)	SVM with rejection	Visual observation	NA	NA
Tang et al. [53]	5 FSRs, ACC.	400	Wireless module	9 adults	23.3 (4.3)	SVM, MLP	Visual observation	Indoor	11.5 h (total)
Zhang et al. [54]	5 FSRs, ACC.	25	Wireless module	9 adults	27.3 (4.3)	DT	Visual observation	NA	11.36 h (total)
Zhang et al. [55]	4 pressure sensors	35	Bluetooth	10 adults	24–56	NA	Visual observation	Community	NA
Chen et al. [56]	4 pressure sensors	250	Wireless module	5 adults, 1 amputee person	23.2 (1.3); 45	DT, LDA	Visual observation	NA	8 h
Cates et al. [57]	4 FSRs, ACC.	20	Bluetooth	20 adults	28 (5)	SVM	Visual observation	NA	2 min/activity
Hegde et al. [58]	3 pressure sensors, ACC.	50	Bluetooth	4 adults	28 (0.5)	MLD	Visual observation	Laboratory	10 min/activity
Cuong Pham et al. [59]	ACC.	50	Wireless module	10 adults	22 (1.7)	CNN	Visual observation	NA	10–30 min/activity
Nguyen et al. [60]	8 pressure sensors, ACC.	50	Bluetooth	3 adults	24–29	DT, KNN, SVM	PPAC and FF + GPS	Indoor and outdoor	15 and 20 m, 17-step (stairs)

ACC.: accelerometer; SVM: Support vector machine; MLP: Multi-layer perceptron; ANN: artificial neural network; LDA: Linear discriminant analysis; KNN: k-nearest-neighbors; MLD: Multinomial Logistic Discrimination; CNN: convolution neural networks; DT: decision tree; NA: not applicable; * Physilog module including an IMU* (accelerometer, gyroscope and magnetometer) and a barometer sensor: Physilog® 10D Silver, GaitUP CH; FF+GPS: foot force sensor and GPS; PPAC: plantar-pressure based ambulatory classification; min: minute; h: hour; FSR: force sensitive resistor.

**Table 3 sensors-19-02438-t003:** Description of technical features and criterion validity of insoles for step count.

	Sensing Elements	Sampling Frequency (Hz)	Data Transmission Methods	Population	Age (years)	Algorithms	Comparison Methods	Settings	Test Durations/Condition	Criterion Validity
Lin et al. [24]	48 textile pressure sensors, IMU*	100	Bluetooth	10 adults	NA	Average method	100 predefined steps	Community	NA	Accuracy: 99.9–100%
Truong et al. [25]	8 pressure sensors, ACC.	50	Bluetooth	7 adults	24.5 (2.14)	Average method	Video record	Indoor	16 m	Accuracy of 100%
Fulk et al. [31]	5 FSRs, ACC.	400	Bluetooth	12 stroke survivors	62.1 (8.2)	Sum method	Video record	Laboratory	2 min	ICC = 0.99
Bakhteri et al. [34]	2 FSRs	NA	Bluetooth	1 athlete	NA	NA	direct observation	Community	720 m	Measurement error: 0%
Ngueleu et al. [35]	5 FSRs	10	Bluetooth	12 adults	21–35	Individual, average and cumulative sum methods	Video record	Indoor and outdoor	6 min	Accuracy: 94.8–99.6%
Rodriguez et al. [36]	1 pressure sensor, ACC.	NA	Wire link	1 adult	NA	NA	Two smartphone applications	NA	50 predefined steps	Error rate of 4% (walking) and 0% (running)
Piau et al. [37]	1 pressure sensor, ACC.	100	Wifi	3 adults	25, 29, 30	Acceleration variance	100 predefined steps	Laboratory	NA	Measurement error: <1%

**Table 4 sensors-19-02438-t004:** Criterion validity of insoles for posture and activity recognition.

	Postures and Activities	Criterion Validity				
		Accuracy ^(a)^	Precision ^(b)^	Specificity ^(c)^	Sensitivity ^(d)^	Identification Rate
Hegde et al. [26]	Lying down, sitting, standing, walking, ascending and descending stairs, vacuuming, shelving items, cycling, washing, sweeping, car driving	81% (overall); 98% (Lying down),88% (sitting), 92% (standing), 96% (walking), 67% (ascending stairs), 41% (descending stairs), 63% (vacuuming), 65% (shelving items), 99% (cycling), 83% (washing), 69% (sweeping), 92% (car driving)				
Fulk et al. [31]	Sitting, standing, walking		From 95.4% to 98.7%		From 95% to 99%	
Achkar et al. [32]	Sitting, standing, walking, elevator up/down, up/downstairs and ascending/descending ramp	From 97.8% to 99.9% *	From 90.3% to 99.1%	From 98.5% to 100%	From 77.7% to 99.6%	
Achkar et al. [33]	Sitting, lying, standing, walking	Accuracy of 93%,	From 91% to 95%	From 93% to 99%	From 88% to 99%	
Anlauff et al. [38]	Standing and walking					92.12% with SD = 6.53 and 66.26% with SD = 15.78 during standing and walking respectively
Chen et al. [39]	Standing, walking, ascending and descending stairs, ascending and descending ramp	99.9% for standing, 98.9% for walking, 99.5% for ascending and 99.5% for descending stairs, 99.1% for ascending and 99.9% for descending ramp				
Fulk et al. [40]	Sitting, standing, walking	95% and 99.9% for group and individual models **			From 82% to 99% and from 99.9% to 100% for group and individual models	
Zhang et al. [41]	Walking, running, ascending and descending stairs					98.3% for testing and 98.8% for training
Zhang et al. [42]	Sitting, standing, walking, ascending/descending stairs, cycling on a stationary bike	91.5% for group and 99.1% for individual (standing, sitting, walking); 80.2% and 97.9% for group and individual models (all activities)	From 82.8% to 97.2 % for group model (standing, sitting, walking); 3% to 92% for group (all activities)		From 82% to 96.8% for group model (standing, sitting, walking); 15% to 93% for group (all activities)	
Edgar et al. [43]	Sitting, standing, walking, ascending and descending stairs, jumping jacks, east coast six count swing dancing	For household activities: 96.67% (sitting), 90% (standing), 100% (walking), 77.67% (ascending stairs) and 95.67% (descending stairs), 96.67% (doing the dishes), 62% (folding laundry and 98.3% (vacuuming), For athletic activities: 96.6% (sitting), 100% (standing), 100% (walking), 100% (standing), 100% (jumping jacks), 79% (skate forward), 96.6% (skate backward), 76% (swing lead), 96.6% swing follow)				
Hegde et al. [44]	Lying down, sitting, standing, walking and cycling	98.3% for SmartStep 3.0 and 98.5% for SmartStep 2.0;	93% and 100% (Lying down), 96% and 100% (sitting), 96% and 100% (standing), 100% (cycling) for SmartStep 3.0 and 2.0;		100% (Lying down), 92% and 97% (sitting), 96% and 100% (standing), 99% and 100% (walking), 100% (cycling) for SmartStep 3.0 and 2.0;	
Lin et al. [45]	Sitting, standing, walking,	100% (sitting), 99.7% (standing), 95.8% (walking),				
Lin et al. [46]	Sitting, standing, walking,	100% (sitting), 99.7% (standing), 95.8% (walking),				
Peng et al. [47]	Sitting, standing, ascending and descending stairs,	92.9% (overall); 100% and 91.5% (standing), 91% and 76.5% (walking), 93.7% and 85% (ascending stairs), 86.7% and 84.5% (descending stairs) for 7 and 4 sensors respectively				
Sazonov et al. [48]	Sitting, standing, walking, jogging, cycling	Overall: 96% obtained by SVM, 95% by MLD and MLP; Recall and precision from 96% to 97% and 97% (sitting), 92% and from 92% to 93% (standing), from 96% to 98% and from 95% to 98% (walking), from 94% to 95% and from 85% to 93% (cycling)	97% (sitting), from 92% to 93% (standing), from 95% to 98% (walking), from 85% to 93% (cycling)		From 96% to 97% (sitting), 92% (standing), from 96% to 98% (walking), from 94% to 95% (cycling)	
Sazonov et al. [49]	Sitting, standing, walking, jogging, cycling, ascending and descending stairs	95.2% ± 3.5% for all sensors, 95.9% ± 3.3% for left shoe and 94% ± 3.1% for right shoe	95% (Sitting), 100% (standing), 99% (walking), 99% (cycling), 78% (ascending stairs),96% (descending stairs)		99% (Sitting), 99% (standing), 99% (walking), 94% (cycling), 90% (ascending stairs), 80% (descending stairs)	
Shang et al. [50]	Sitting, standing, walking, falling down					100% (standing), from 0% to 98% (sitting), from 97% to 99% (walking), 100% (falling down)
Sugimoto et al. [51]	Sitting, standing, walking, running, ascending and descending stairs, cycling	From 85% to 90%				
Tang et al. [52]	Sitting, standing, walking, running	92.4% and 99.2% without and with linear kernel; 97% and 99.1% without and with RBF kernel;		From 91.3% to 99.8% (sitting), from 96% to 100% (standing), from 96.4% to 99.6% (walking), from 43.1% to 97.4% (ascending stairs), from 50.2% to 87.2% (descending stairs), from 91.2 to 99.2% (cycling) using without and with RBF and linear kernels	From 84.1 to 98.1 (sitting), from 95.9% to 99.9% (standing), from 96.8 to 100% (walking), from 53.7 to 90.5% (ascending stairs) from 46.8 to 91.7 (descending stairs), from 95.5% to 99.9% (cycling) using without and with RBF and linear kernels	
Tang et al. [53]	Sitting, standing, walking, jogging, ascending and descending stairs, cycling	97% and 78% with SVM, 98.7% and 95.9% with SVM_rej, 97.3% and 96.1% with MLP, 99.8% and 98% with MLP_rej on raw and feature data respectively.	from 66.6% to 97.6% (sitting), from 65.8% to 99.9% (standing), from 94.4% to 100% (walking), from 27.2% to 92% (ascending stairs), from 20.9% to 98.3% (descending stairs), from 89.9% to 100% (cycling) without and with rejection on raw and feature data using SVM		From 77.1% to 99.9% (sitting), from 75.6 to 100% (standing), from 83% to 99.9% (walking), from 17.5% to 99.3% (ascending stairs), from 10% to 98.3% (descending stairs), from 81.1% to 99.6% (cycling) without and with rejection on raw and feature data using SVM.	
Zhang et al. [54]	Sitting, standing, walking, jogging, ascending stairs, descending stairs, cycling	98.85% without boosting and 98.90% after boosting algorithm				
Zhang et al. [55]	walking, cycling, bus passenger, car passenger, and car driver	75% (with 2 sensors per foot), 91% (with 4 sensors) and 93% (with 6 sensors)				
Chen et al. [56]	Sitting, standing, walking, obstacle clearance, ascending/descending stair	Overall: 98.8% ± 0.5% and 98.4% (healthy and amputee people); 99.8% ± 0.1% and 100% for sitting, 99.8% ± 0.1% and 100% for standing, 98.7% ± 1% and 98.4% for walking, 97.9% ± 0.6% and 96.8% for obstacle clearance, 98.5% ± 0.9% and 98.1% for ascending stairs, 97.6% ± 1% and 96.9% for descending stairs respectively.				
Cates et al. [57]	Sitting, lying, standing, walking, running, ascending/descending stair, jumping	Sitting (99.7%), lying (97.7%), standing (98.5%), walking (97.8%), running (98.3%), ascending (97%)/descending (96.8%) stair, jumping (99.3%)		Sitting (99.7%), lying (98.3%), standing (99.5%), walking (99.2%), running (98.7%), ascending (98.1%)/descending (98.2%) stair, jumping (99.9%)	Sitting (85.8%), lying (99.3%), standing (92.3%), walking (95.5%), running (90.5%), ascending (87.3%)/descending (85%) stair, jumping (93.8%)	
Hegde et al. [58]	Sitting, standing, walking, cycling	96.6% (overall); more than 99% for sitting and standing; less of 90% for cycling				
Cuong Pham et al. [59]	Standing, running, walking, cycling, jumping, kicking		93.4% (overall); 88.3% or standing, 85.4% for running, 100% for walking, 95.1% for cycling, 97.3% for jumping, 94.4% for kicking		93.2% (overall); 87.2% for standing, 97.4 for running, 97.4% for walking, 100% for cycling, 91% for jumping, 85.9% for kicking	
Nguyen et al. [60]	Level ground ascending/descending stair, ascending/descending incline	97.84% (overall); 98.11% (level ground), 98.11 (stair descent), 99.73% (stair ascent), 100% (incline descent), 99.73% (incline ascent)	91.35% (level ground), 98.64% (stair descent), 100% (stair ascent), 100% (incline descent), 100% (incline ascent)		100% (level ground), 100% (stair descent), 98.65% (stair ascent), 100% (incline descent), 90.54% (incline ascent)	

$\left(a\right)\;\mathrm{Accuracy}=\frac{\mathrm{true}\;\mathrm{positive}\;}{\mathrm{true}\;\mathrm{positive}\;+\;\mathrm{false}\;\mathrm{positive}\;+\;\mathrm{false}\;\mathrm{negative}}$ except in [32] where $*\;\mathrm{accuracy}=\frac{\mathrm{true}\;\mathrm{positive}\;+\;\mathrm{true}\;\mathrm{negative}}{\mathrm{total}\;\mathrm{sample}\;\mathrm{number}}$; $\left(b\right)\;\mathrm{Precision}=\frac{\mathrm{true}\;\mathrm{positive}\;}{\mathrm{true}\;\mathrm{positive}+\mathrm{false}\;\mathrm{positive}}$; $\left(c\right)\;\mathrm{Specificity}=\;\frac{\mathrm{true}\;\mathrm{negative}}{\mathrm{true}\;\mathrm{negative}+\mathrm{false}\;\mathrm{positive}}$; $\left(d\right)\;\mathrm{Sensitivity}=\frac{\mathrm{true}\;\mathrm{positive}}{\mathrm{true}\;\mathrm{positive}\;+\;\mathrm{false}\;\mathrm{negative}}$. ** For individual models, a classifier was trained for each individual participant. The group model was trained on the data pooled from several participants.

**Table 5 sensors-19-02438-t005:** Summary of methodological quality appraisal of included studies using MacDermid criteria.

Study References	MacDermid Criteria												Total	Overall Score (%)	Quality Score
	1	2	3	4	5	6	7	8	9	10	11	12			
[31]	2	2	2	2	1	-	2	2	2	2	-	2	19	95%	HQ
[38]	2	2	2	2	1	-	2	1	2	2	-	2	18	90%	HQ
[34]	2	2	2	2	1	-	2	1	2	2	-	2	18	90%	HQ
[26]	2	2	2	2	1	-	2	1	2	2	-	2	18	90%	HQ
[48]	2	2	2	2	1	-	2	1	2	2	-	2	18	90%	HQ
[47]	2	2	2	2	1	-	2	1	2	2	-	2	18	90%	HQ
[52]	2	2	2	2	1	-	2	1	2	2	-	2	18	90%	HQ
[53]	2	2	2	2	1	-	2	1	2	2	-	2	18	90%	HQ
[41]	2	2	2	2	1	-	2	2	2	2	-	1	18	90%	HQ
[35]	2	2	2	1	1	-	2	2	2	2	-	2	18	90%	HQ
[42]	2	2	2	2	0	-	2	1	2	2	-	2	17	85%	HQ
[44]	2	1	2	2	1	-	2	1	2	2	-	2	17	85%	HQ
[32]	2	2	2	2	1	-	2	1	2	2	-	1	17	85%	HQ
[46]	2	2	2	2	0	-	2	1	2	2	-	2	17	85%	HQ
[56]	2	2	1	2	1	-	2	1	2	2	-	2	17	85%	HQ
[37]	1	2	2	2	1	-	2	1	2	1	-	2	16	80%	HQ
[39]	2	2	2	2	1	-	2	1	2	2	-	2	16	80%	HQ
[43]	1	2	2	2	0	-	2	1	2	2	-	2	16	80%	HQ
[45]	1	2	2	2	0	-	2	1	2	2	-	2	16	80%	HQ
[51]	1	2	2	2	1	-	1	2	2	2	-	1	16	80%	HQ
[40]	1	2	1	2	1	-	2	1	2	2	-	2	16	80%	HQ
[55]	2	2	1	2	1	-	2	1	2	2	-	2	16	80%	HQ
[24]	2	2	2	2	1	-	1	0	1	2	-	2	15	75%	GQ
[49]	2	1	2	2	0	-	2	1	2	2	-	1	15	75%	GQ
[59]	2	1	2	2	0	-	1	1	2	2	-	2	15	75%	GQ
[54]	2	0	2	2	0	-	2	1	2	2	-	1	14	70%	GQ
[36]	1	1	2	2	0	-	1	1	2	2	-	2	14	70%	GQ
[25]	2	2	2	2	0	-	1	1	2	1	-	1	14	70%	GQ
[57]	2	1	2	2	0	-	1	1	2	1	-	2	14	70%	GQ
[35]	1	1	2	2	0	-	1	1	2	2	-	2	14	70%	GQ
[50]	1	1	2	2	0	-	1	1	1	2	-	1	12	60%	MQ
[58]	2	2	2	1	0	-	1	1	1	1	-	1	12	60%	MQ
[33]	2	1	1	1	0	-	2	1	1	0	-	1	10	50%	MQ

High quality” (HQ) ≥ 80.0%, “good quality” (GQ) between 70.0% and 79.9%, “moderate quality” (MQ) for scores between 50.0% and 69.9%, and “low quality” (LQ) < 50%. MacDermid criteria [29]: **1**. Was the relevant background research cited to define what is currently known about the psychometric properties of the measures under study, and the need or potential contributions of the current research question? **2**. Were appropriate inclusion/exclusion criteria defined? **3**. Were specific psychometric hypotheses identified? **4**. Was an appropriate scope of psychometric properties considered? **5**. Was an appropriate sample size used? **6**. Was appropriate retention/follow-up obtained? (Studies involving retesting or follow-up only) **7**. Documentation: Were specific descriptions provided or referenced that explain the measures and its correct application/interpretation (to a standard that would allow replication)? **8**. Standardized Methods: Were administration and application of measurement techniques within the study standardized and did they are considered potential sources of error/misinterpretation? **9**. Were analyses conducted for each specific hypothesis or purpose? **10**. Were appropriate statistical tests conducted to obtain point estimates of the psychometric property? **11**. Were appropriate ancillary analyses were done to describe properties beyond the point estimates (Confidence intervals, benchmark comparisons, SEM/MID)? **12**. Were the conclusions/clinical recommendations supported by the study objectives, analysis, and results?

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
