# Peer review of "Validity of Instrumented Insoles for Step Counting, Posture and Activity Recognition: A Systematic Review"

_sensors, 2019, doi:10.3390/s19112438_

Round 1

Reviewer 1 Report

Thank you for allowing me to review this paper - it is an interesting systematic review to assess and synthesise the available information on the criterion validity of the instrumented insoles in identifying posture. Here are my comments:

Methods:
-    Although five electronic databases (MEDLINE, EMBASE, IEEE Xplore, Cochrane Library and Web of Sciences) were searched for studies reporting, I suggest exploring the European database - Scopus.
- Here we would like to suggest adding the newest publication, which is in the topic of the manuscript: Michal Ostaszewski, Jolanta Pauk: Estimation of ground reaction forces and joint moments on the basis on plantar pressure insoles and wearable sensors for joint angle measurement, Technol. a. Health Care 2018, 26, 2: 605-612.
https://www.researchgate.net/publication/325306505_Estimation_of_ground_reaction_forces_and_joint_moments_on_the_basis_on_plantar_pressure_insoles_and_wearable_sensors_for_joint_angle_measurement

Author Response

We thank all reviewers for their contributive comments. Here below are our point by point responses to reviewer 1 comments (also provided in the attached file).

Reviewer #1:

Point 1: Thank you for allowing me to review this paper - it is an interesting systematic review to assess and synthesise the available information on the criterion validity of the instrumented insoles in identifying posture. Here are my comments:

Methods:
-    Although five electronic databases (MEDLINE, EMBASE, IEEE Xplore, Cochrane Library and Web of Sciences) were searched for studies reporting, I suggest exploring the European database – Scopus

Response: We thank the reviewer for this critical and constructive comment.

We have updated the bibliography search, adding Scopus to electronic databases. Only one more article was found to meet the inclusion criteria. This article has now been added to our systematic review, and extracted results are presented in Table 2.

Point 2: Here we would like to suggest adding the newest publication, which is in the topic of the manuscript: Michal Ostaszewski, Jolanta Pauk: Estimation of ground reaction forces and joint moments on the basis on plantar pressure insoles and wearable sensors for joint angle measurement, Technol. a. Health Care 2018, 26, 2: 605-612.
https://www.researchgate.net/publication/325306505_Estimation_of_ground_reaction_forces_and_joint_moments_on_the_basis_on_plantar_pressure_insoles_and_wearable_sensors_for_joint_angle_measurement

Response 2: Thank you for this interesting reference that we have examined carefully. However, it appears that this study lies beyond the scope our systematic review. Indeed, the present systematic review focuses on the validity of instrumented insoles for step count or posture and activity recognition. Even though the study by Ostaszewski et al (2018) presents data related to plantar pressure insoles and wearable sensors, their two main variables of interest were ground reaction forces and joint moments. The article reported no data related to step count or posture and activity recognition, and therefore cannot be included in the systematic review.

Reviewer 2 Report

This literature review is dedicated to instrumented shoe inserts for activity monitoring. This is a timely topic that has not previously been the subject of a review article. The manuscript is well written, has an easy to follow structure, and covers all the important aspects of the topic.

Minor revisions should address some typos and awkward phrasing (Lines 63, 67, 75, 145, 271, 304, ...) Other issues to consider are listed below.

L 79: This listing of measurement properties under “validity” may be contested as conflating different concepts (e.g., Criterion validity is not the same sort of property as sensitivity or measurement error. Rather it is a sub-category of validity that is defined by such things as sensitivity and error rate, as spelled out on line 105.)

L 95: The inception date of the database appears like an arbitrary cut-off point, unless it is (wrongly) assumed that no articles with an earlier publication date are included in these databases.

L 109: Does that mean that all those variables were computed by the reviewers based on the raw data, or were reported numbers taken at face value? If not, were studies excluded that did not report their methods of calculating these variables?

L 134: Criterion validity should not be listed as a sub-category of criterion validity

L 159: The chart title of Figure 2 should be removed (this information is already in the caption)

L 217: This sentence may be confusing and could be struck or moved to the beginning of the following chapter

L 248: How does the variable “identification rate” relate to anything mentioned in the Methods section? If this is not translatable into any of the specified validity variables it may be necessary to introduce it as a separate variable earlier.

L 274: Can “monitoring” be specified?

L 276: This statement could be split up to better distinguish between results that are attributable to the device being tested and those that are affected by limitations of the respective test protocols.

L 294: It may be worth mentioning that, in absence of an all-around “best” device, the findings may help identify the most appropriate device for a given application.

L 307: “Less accurate” than what?

Author Response

We thank all reviewers for their contributive comments. Here below are our point by point responses to reviewer 2 comments (also provided in the attached file).

Reviewer 2:

This literature review is dedicated to instrumented shoe inserts for activity monitoring. This is a timely topic that has not previously been the subject of a review article. The manuscript is well written, has an easy to follow structure, and covers all the important aspects of the topic.

Point 1: Minor revisions should address some typos and awkward phrasing (Lines 63, 67, 75, 145, 271, 304, ...) Other issues to consider are listed below.

Response 1: Changes have been made accordingly throughout the manuscript, including in lines 63, 67, 75, 145, 271 and 304

Point 2: L 79: This listing of measurement properties under “validity” may be contested as conflating different concepts (e.g., Criterion validity is not the same sort of property as sensitivity or measurement error. Rather it is a sub-category of validity that is defined by such things as sensitivity and error rate, as spelled out on line 105.)

Response 2: We thank the reviewer for this comment. To clarify this point, we have reworded the portion where measurement properties where listed. Modifications are can highlighted in the manuscript (page 3, lines 107-109).

Point 3: L 95: The inception date of the database appears like an arbitrary cut-off point, unless it is (wrongly) assumed that no articles with an earlier publication date are included in these databases.

Response 3: The consideration of inception date of the databases allowed us to include all articles that were accessible in these databases, regardless of the articles dates of publication if they are indexed in those databases.

Point 4: L 109: Does that mean that all those variables were computed by the reviewers based on the raw data, or were reported numbers taken at face value? If not, were studies excluded that did not report their methods of calculating these variables?

Response 4: Reported numbers were taken at face value, as published in the articles. The methods of calculation we presented in the systematic review were also extracted from the articles that reported their methods.

Point 5: L 134: Criterion validity should not be listed as a sub-category of criterion validity

Response 5: Thank you for this comment. We agree that criterion validity should not be listed twice in the sentence. The sentence has been revised accordingly (page 4; line 135)

Point 6: L 159: The chart title of Figure 2 should be removed (this information is already in the caption)

Response 6: Figure 2 has been updated to reflect the most recent bibliography search. The chart title has accordingly been removed as requested (page 6)

Point 7: L 217: This sentence may be confusing and could be struck or moved to the beginning of the following chapter

Response 7: Thank you for this suggestion. To clarify, we have removed the confusing sentence.

Point 8: L 248: How does the variable “identification rate” relate to anything mentioned in the Methods section? If this is not translatable into any of the specified validity variables it may be necessary to introduce it as a separate variable earlier.

Response 8: “identification rate” relates to the accuracy of detection.

Point 9: L 274: Can “monitoring” be specified?

Response 9: in this sentence, we were referring to the monitoring of steps. In the revised manuscript, the sentence now reas as follows: “monitoring of steps”

Point 10: L 276: This statement could be split up to better distinguish between results that are attributable to the device being tested and those that are affected by limitations of the respective test protocols.

Response 10: Following the above comment, we have reworded the sentence to reflect more clearly results that are attributable to devices and those related to test protocols (page 22; line 285).

Point 11: L 294: It may be worth mentioning that, in absence of an all-around “best” device, the findings may help identify the most appropriate device for a given application.

Response 11: We thank the reviewer for this relevant suggestion. Change have been made accordingly in the manuscript (page 22; lines 306-307).

Point 12: L 307: “Less accurate” than what?

Response 12: Change has been made by adding “than other types of activity monitoring.” in sentence. Revised sentence is as follows (page 23; lines 318-319):

 “Therefore, more studies are required to investigate the accuracy of instrumented insoles in discriminating standing from walking at a slow speed.”

Reviewer 3 Report

In the manuscript, 32 papers were selected and analyzed among a lot of insole-related papers. From the systematic review, readers may easily grasp the insight into the validity of instrumented insoles in detecting postures activities and steps. This paper is a well-organized paper with in-depth analysis. I believe that this contributes to the related field. One minor comment is that the rows in Table 4 are not clearly-divided. Revision is required on this matter for better readerbility.

Author Response

We thank all reviewers for their contributive comments. Here below are our point by point responses to reviewer 3 comments.

Reviewer #3:

Point: In the manuscript, 32 papers were selected and analyzed among a lot of insole-related papers. From the systematic review, readers may easily grasp the insight into the validity of instrumented insoles in detecting postures activities and steps. This paper is a well-organized paper with in-depth analysis. I believe that this contributes to the related field. One minor comment is that the rows in Table 4 are not clearly-divided. Revision is required on this matter for better readerbility.

Response: We thank the reviewer for this critical and constructive comment. Table 4 has been adapted in the revised version of manuscript. Divisions within the table are now clear.

Round 2

Reviewer 1 Report

All corrections have been successfully provided

Author Response

We thank the reviewer for his critical and helpful comments during the first round of review.

Reviewer 2 Report

My previous comments have been sufficiently addressed.

Author Response

(The authors gave the same response as above.)

Reviewer 3 Report

The author fairly addressed the reviewer's comments. The paper can be accepted in its current form.

Author Response

(The authors gave the same response as above.)
